# Skin-electrode iontronic interface for mechanosensing

Pang Zhu[1,5], Huifeng Du [2,5], Xingyu Hou[1,5], Peng Lu[1], Liu Wang [1,2,3], Jun Huang[1], Ningning Bai[1], Zhigang Wu[1,4], Nicholas X. Fang[2✉] & Chuan Fei Guo [1,3✉]

Electrodermal devices that capture the physiological response of skin are crucial for monitoring vital signals, but they often require convoluted layered designs with either electronic or ionic active materials relying on complicated synthesis procedures, encapsulation, and packaging techniques. Here, we report that the ionic transport in living systems can provide a simple mode of iontronic sensing and bypass the need of artificial ionic materials. A simple skin-electrode mechanosensing structure (SEMS) is constructed, exhibiting high pressure-resolution and spatial-resolution, being capable of feeling touch and detecting weak physiological signals such as fingertip pulse under different skin humidity. Our mechanical analysis reveals the critical role of instability in high-aspect-ratio microstructures on sensing. We further demonstrate pressure mapping with millimeter-spatial-resolution using a fully textile SEMS-based glove. The simplicity and reliability of SEMS hold great promise of diverse healthcare applications, such as pulse detection and recovering the sensory capability in patients with tactile dysfunction.

[1] Department of Materials Science and Engineering, Southern University of Science and Technology, Shenzhen, Guangdong, China. [2] Department of Mechanical Engineering, Massachusetts Institute of Technology, Cambridge, MA, USA. [3] Centers for Mechanical Engineering Research and Education at MIT and SUSTech, Southern University of Science and Technology, Shenzhen, Guangdong, China. [4] State Key Laboratory of Digital Manufacturing Equipment and Technology, Huazhong University of Science and Technology, Wuhan, China. [5] These authors contributed equally: Pang Zhu, Huifeng Du, Xingyu Hou. ✉email: nicfang@mit.edu; guocf@sustech.edu.cn

The richness of sensory function of skin is made possible with a large variety of receptors to provide afferent information related to touch (mechanoreceptors), pain (nociceptors), and temperature (thermoreceptors)[1–3]. Recently, a great need of monitoring vital signals of the human body[4–6], as well as the precise measurement of finger or hand manipulation under either external or internal stimuli has promoted the scientific and technological breakthroughs on wearable and epidermal electronic sensors[7,8]. To mimic the dynamic and micromechanical sensory function of skins, the current form of electronic skins often takes a layered structure of two electrodes sandwiching a piezoresistive[9], dielectric[7,8,10–12], or piezoelectric layer[13]. However, these electronic skins often suffer from sophisticated material synthesis protocols and the need of extra encapsulation to maintain the hydrated functional environment. For example, Xia et al. reported a capacitive-type electronic skin, of which a layer of microgels is used as the deformation component and two layers of Ag-coated PDMS films serve as electrodes. The multilayer structure is sealed using tapes before it is attached on the human skin to detect physiological signals[14]. In addition, the accuracy of recording such vital signal with desired spatial and temporal resolution might be significantly compromised by the presence of an epidermal barrier consisting of dead cell material (the stratum corneum). Electrical noise is a challenge of high-quality signal recording for epidermal and wearable electronic sensors, and is dominantly attributed to the relative motion at the skin-electrode interfaces[15]. Although most wearables such as smartwatches can display real-time heart rates using photoplethysmographic (PPG) sensors that detect changes in tissue blood volume using a photodetector[16], the accuracy of such devices is subject to noise introduced by the variation of skin contact on the patient or device movement, environmental conditions, and ectopic beats[17], therefore using wearable devices for the analysis of heart rate anomaly is often limited.

On the other hand, the skin keeps naturally wet thanks to the presence of the sweat glands and the porous nature of the stratum corneum. Under physiological conditions, the stratum corneum is always partially hydrated. The sweat ducts function as pumps to pour sweats toward the skin surface, and the sweat penetrates into the porous corneum as a result of high pressure in the duct and/or diffusion[18]. Since sweat contains numerous ions such as $Na^+$ and $Cl^-$ on the order of 10 mM in concentration[19], the ionic conductance of the corneum increases when being soaked with sweat. As a result, when we put a metal electrode in conformal contact with the skin, free electrons serve as the carriers in the electrodes while ionic fluxes contribute to the conduction in the tissue to exchange electronic and ionic signals[20]. This is a natural iontronic interface, since the ions involved in the current flow through ductal sweat and interstitial fluid are accumulating at these boundaries, followed by the buildup of a potential difference across the cell membrane, the direction of which is opposite to the applied voltage—hence called "counter electromotive force".

Laminating electrodes on skin for collecting bioelectricity signals, such as electrodermal response and electrocardiographic (ECG) signals has been widely used in healthcare for over 100 years[21]. The electrode-skin interface is of iontronic nature. For an iontronic interface, electrons and ions are separated without electrochemical reaction when applied with a small voltage[22], and the capacitance is proportional to the interfacial contact area between the electrode and the electrolyte[23–26]. Whereas such an "electrodes-on-skin" configuration is widely applied in electrodermal signal recording and stimulation, to our knowledge, it has not been used in tactile sensing despite its huge potential. The naturally possessed internal ion-exchanging capability and waterproof protective enclosure of epidermis make human skin

an ideal selection of overcoming the giant challenges in implementing synthetic conductive gel-based sensors.

In this work, we utilized the ionic transport in living systems to construct an iontronic sensing structure: skin-electrode mechanosensing structure (SEMS), which simply consists of two electrodes and the skin. It exhibited high pressure resolution and spatial resolution, being capable of feeling touch and detecting weak physiological signals such as fingertip pulse under different skin humidity. We studied the influence of motion artifacts on the detection of pulse signals. We were delighted to find that motion-related frequencies can be conveniently distinguished from the characteristic frequencies of pulse in the frequency domain, thus the extraction of physiological signal can be finished without being affected by motion artifacts. We further fabricated a fully textile SEMS-based glove for pressure mapping with millimeter-spatial resolution. The simplicity and reliability of the SEMS hold much potential in a wide range of healthcare applications, such as pulse detection in daily life and helping the patients with tactile dysfunction to recover sensory capability.

## Results

**Principle, structure, and performance of the skin-electrode mechanosensing structure.** The apparent contrast between ECG (mainly an internal stimulation) and tactile sensing to grasped objects (a response of the skin to external mechanical stimulation) can be elaborated by their underlying working principles. For measurements of epidermal conductance and ECG, electrodes are required to maintain an intimate and conformable contact with the skin to ensure stable potential signal recording (Supplementary Fig. 1a). For sensing, however, a tunable interfacial contact or capacitance signal that is sensitive to mechanical stimuli is expected (Supplementary Fig. 1b). These crucial yet contradictory requirements necessitate a comprehensive strategy of the design, behavioral prediction, and fabrication of skin-interfaced pressure sensing structures.

Here we demonstrate a SEMS that utilizes the skin as an ionic material for measuring both physiological signals and external mechanical stimuli. A SEMS simply consists of the skin, a sensing electrode (SE) with microstructured surfaces that allow for subtle change in skin-electrode contact, and a highly conformable counter electrode (CE) (Fig. 1a). These soft electrodes are effortlessly fixated on skin using a piece of transparent and breathable healthcare film (details are seen in "Methods"), which applies a base-pressure of ~5 kPa on the SEMS. In contrast with complicated design of existing e-skins which often have a layered structure, the SEMS has a much simpler structure with electrodes laminated on the skin surface in parallel without employing any synthesized ionic gels or hydrogels. Moreover, on top of the high specific capacitance of the iontronic interface, we provide physical insights into the important geometric features that govern the instability-enabled pressure sensitivity increase from mechanical analysis and simulation, which accurately predicts the deformation response and guides the design of microstructures on the electrodes.

To illustrate our concept of SEMS, we fabricated Au-coated polydimethylsiloxane (PDMS) micropillars (Fig. 1b) with a length ($L$) to radius ($R$) aspect ratio of 6 as the SE. PDMS is a soft material whose biocompatibility and breathability have been repeatedly confirmed by in vitro and in vivo studies[27,28]. Upon loading, the soft micropillared structures allow a sensitive change in contact area of the skin-electrode interface (Fig. 1c), which behaves like a capacitor with a much higher unit area capacitance than conventional capacitors. The SE also presents desired biocompatibility and breathability as an on-skin adhesive patch because of the minimized contact with the skin, as well as the

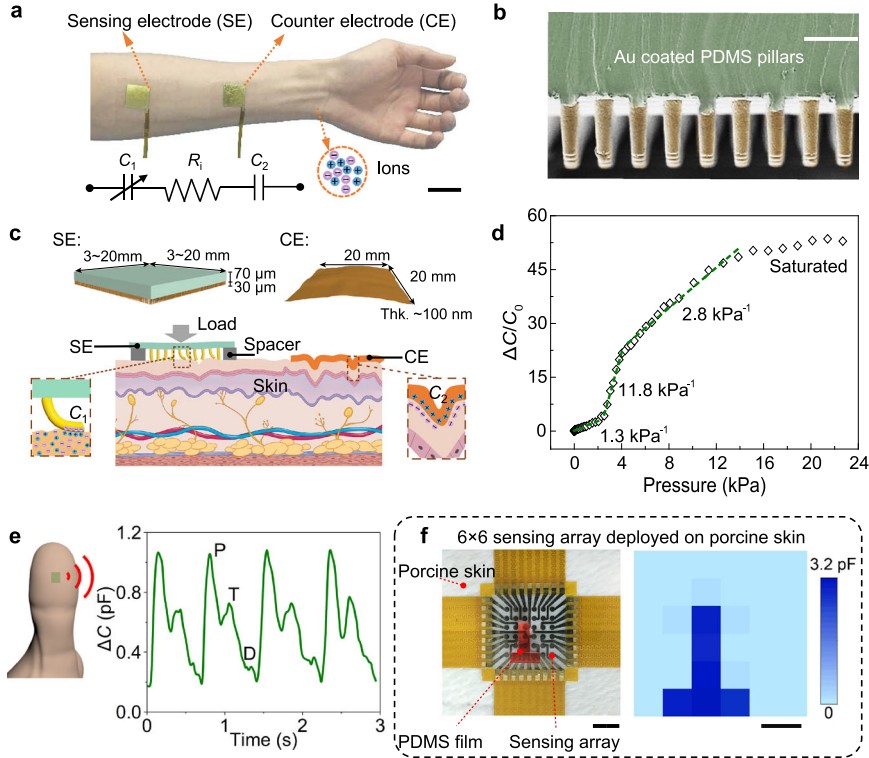

**Fig. 1 Working mechanism and sensing performance of the SEMS. a** Photograph of the SEMS with a sensing electrode (SE) and a counter electrode (CE) laminated on an arm, and a simplified equivalent circuit of the SEMS. Scale bar: 5 cm. **b** Scanning electron microscopy (SEM) image of the micropillared SE. Scale bar: 50 μm. Similar result can be repeated for at least three times. **c** Exploded view and schematic illustration of the SEMS. The insets of the schematic show the skin-electrode interface upon loading. **d** Normalized change in capacitance as a function of applied pressure of the SEMS with a spacer measured at $10^4$ Hz. **e** Typical fingertip pulse waveform measured using the SEMS. **f** A 6 × 6 SEMS pixel array that loads a T-shaped PDMS stamp (scale bar: 2 cm) and corresponding mapping of the signal (scale bar: 1 cm).

gaps between the pillars that help ventilate air and skin residues to keep the skin dry[29,30], although PDMS is airtight and watertight. The CE is a piece of thin Au film leaf or Au nanomesh that is fully conformable to the skin texture such that the corresponding iontronic capacitance is constant. The equivalent circuit of the SEMS can be simplified as two capacitors ($C_1$ for SE; and $C_2$ for CE, which has a fixed value determined to be $\sim 100 \, \text{nF·cm}^{-2}$) and a resistor ($R_i$, stands for the resistance between the two electrodes) in series, and then in parallel with the electrode coupling capacitance $C_E$, which is small and can be negligible. A rough estimation of the measured capacitance $C$ follows $1/C = 1/C_1 + 1/C_2$. Since $C_2$ is a constant, the signal output is determined by $C_1$, which is a function of applied pressure. The distance between the two electrodes has little influence on $C$ because of the ionic nature of the skin (Supplementary Fig. 2a), which can be verified by the fact that capacitance significantly decreases with increasing test frequency —a characteristic of iontronic devices (Supplementary Fig. 2b). Such a feature greatly simplifies the deployment of the electrodes in the SEMS, and allows a set of SEs to share a single CE for sensing array applications.

As a key parameter for pressure sensing of the SEMS, the sensitivity is defined as $S = \delta(\Delta C/C_0)/\delta P$, where $C_0$ is the capacitance before loading, $\Delta C$ is the change in capacitance, and $P$ is the applied pressure. Upon loading, the micropillars bend down to make an intimate contact with the skin, resulting in an increase of capacitance. From the definition of sensitivity, decreasing $C_0$ or initial contact area will lead to the increasing of sensitivity. A spacer, which is a perforated polyethylene terephthalate (PET) membrane (details are described in Supplementary Fig. 3), was therefore placed between the SE and skin to

minimize initial contact area and thereby improve sensitivity on the one hand, and improve the consistency between different devices on the other hand. The contact impedance values between electrodes and skin of both CE and SE (with and without a spacer) are within $10^2 - 10^5 \, \Omega$ in a frequency range of $10^4 - 10^6$ Hz. At $10^5$ Hz, the mostly used frequency in our study, the impedance is several thousand ohms (Supplementary Fig. 4). The SEMS exhibits nonlinear capacitance-to-pressure response that can be divided into four stages: sensitivity is $\sim 1.3 \, \text{kPa}^{-1}$ under $P < 3$ kPa, and dramatically increases to $11.8 \, \text{kPa}^{-1}$ within 3–4 kPa, and then drops to $2.8 \, \text{kPa}^{-1}$ from 4 to 15 kPa (Fig. 1d). Finally, the response gets saturated as pressure goes beyond 15 kPa. Limit of detection (LOD) of the SEMS is determined to be $\sim 0.2$ Pa and response time is determined to be $\sim 15$ ms (Supplementary Fig. 5), which are superior to that of the human skin (with a LOD of $\sim 100$ Pa and a response speed of 30–50 ms)[2]. We need to point out that the spacer can be removed for more comfortable attachment, and the maximum sensitivity of the SEMS without a spacer is $3.1 \, \text{kPa}^{-1}$ (Supplementary Fig. 6). We compared the performances of the SEMS with a few other e-skins which are used for tactile and physiological sensing. Our SEMS presented higher sensitivity compared with most capacitive-type e-skins but lower than most iontronic e-skins (Supplementary Table 1) because the stratum corneum layer on the skin surface significantly compromises the sensitivity.

The SEMS can sensitively respond to mechanical stimuli of either external forces or physiological information from the body, exemplified by the successful detection of gentle touches with increased force (Supplementary Fig. 7), and blood pulse waveform of a fingertip with clear characteristic P-, T-, and D-waves[31], as well as low noise for each pulse (Fig. 1e). Note that the fingertip

pulse causes a tiny pressure variation of only ~10 Pa which is far weaker than the radial artery pulse, and thus can hardly be detected using existing e-skins or by the human skin for which pressure resolution is only ~7%[32]. By contrast, our SEMS exhibits an extremely high pressure resolution of ~1 Pa, or ~0.02% at a base-pressure of ~5 kPa, and is able to detect the feature of the P-, T-, and D-waves with sharp peaks. The pulse is usually detected from the radial artery, but the concave surface of the wrist causes unstable signal during hand motion. By contrast, the convex surface of fingertips guarantees the signal stability even when the finger is moving.

A SEMS pixel array is further demonstrated in Fig. 1f for pressure mapping. The pixel array is 1.87 cm × 1.87 cm in area consisting of 36 circular SEs with a diameter of 1.7 mm and one shared CE. The capacitance distribution recorded by the pixel array precisely reflects the applied pressure from a piece of "T" shaped PDMS stamp. In addition, our result shows that the SEMS is capable of resolving pressure at a submillimeter spatial resolution (Supplementary Fig. 8) while exhibiting high signal-to-noise ratio and negligible crosstalk between neighboring sensing elements.

**Sensing mechanism of the SEMS based on finite element analysis.** Because the iontronic capacitance is proportional to the skin-electrode contact area, we harness the buckling instability of the slender micropillars to maximize the change in contact area under external load. As illustrated in Fig. 2a, a deformed pillar in a compression test may take different morphologies depending on the length to radius aspect ratio ($L/R$) and its stiffness relative to the substrate. In pre-buckling configuration, a pillar with a semi-spherical end is pressed into the skin, while in the post-buckling regime the pillar is bent over with its axis parallel to the substrate and side area compressed against an impinging elastic plane. Both the pre- and post-buckling contact area can be calculated based on contact mechanics models, and the area change around the onset point of Euler buckling[33] with respect to different combinations of two parameters is plotted in a color map. Since the most rapid change in contact area (or the maximum sensitivity) happens right after the initiation of buckling instability with a larger aspect ratio we expect to have a more drastic area change per unit load, corresponding to a higher sensitivity (Fig. 2b). Therefore, electrodes with slender beams or other hairy structures will be highly favorable for the construction of supercapacitive sensors.

We further validated the experimental results of the microstructured electrode with finite elemental analysis (FEA). Following the previous classification, the normalized change in area of pillars with different aspect ratios corresponds well to different slops of the normalized contact area-pressure plot, and the curve agrees remarkably well with the sensitivity results. The pre-buckling, transition, post-buckling, and saturated phases have also been observed in our scanning electron microscopy (SEM) inspection, and the simulated shapes are highly consistent with the SEM images of different stages, as shown in Fig. 2c.

**Stability, skin irritation, and interference immunity of the SEMS.** Epidermal electronics are required to be stable during signal recording in either static or dynamic states, and be safe to the skin during long-term wearing[29,34,35]. Here both the epidermal SE and CE exhibit high flexibility, without generating cracks upon squeezing (Supplementary Fig. 9) or stretching to a strain of ~10% (Fig. 3a and b). The CE is a thin gold leaf that fully conforms to the skin texture, as such the capacitance signal of the SEMS keeps almost unchanged when touching the CE (Fig. 3c and Supplementary Movie 1). The capacitance signal of the SE

may vary at different sites of human body including wrist, fingers, palm, finger joints, and forearm (Supplementary Fig. 10). At these sites, the skin serves as a surface of different curvatures and different hydration conditions. On the other hand, the elastic microstructures of the SE enable high reliability in signal measurement as well: the response of the SEMS remains highly stable after 5000 cycles of repeated loading (5 kPa)-unloading (Fig. 3d). Although cracks are generated in the gold layer coated on the PDMS pillars when subjected to large deformations because of the huge difference in elastic moduli between the gold film and PDMS, the formation of delocalized cracks, however, does not cause significant change in electrical conduction[36,37]. This is further verified by the bending test over 5000 cycles with a bending radius of 0.6 cm (Supplementary Fig. 11), showing that the resistance of the electrode has little change over cycling.

In addition, both signal stability and skin irritation are evaluated during attachment of the SEMS for a whole day. Acceptable change in capacitance signal was observed when the SEMS is attached on the skin for 24 h (Supplementary Fig. 12); and all six subjects involved in skin safety test did not report discomfort during 24 h wearing of the electrodes, and no skin irritation or redness was reported in the sensing area (Fig. 3e and f). The micropillared electrode was also safely laminated on the skin of a subject for 10 days, but more serious redness was observed in the tape-covered area.

The capacitive nature of our SEMS makes it suitable for recording over a wide spectrum of AC frequencies and/or the use of transients, and it opens new avenue to quantitatively comprehend the ionic properties of the electrodermal system, with new insights on their origins and interaction. For example, the skin regulates emission of sweat through a combined pumping and evaporation through the skin, and such perspiration process plays an important role in thermal comfort[38]. Such dynamic sweating process might affect the skin-electrode interface, and its effect on long-term stability of pulse and tactile sensing should be clarified. We conducted two experiments to validate the effect of skin humidity, and the size of sensing electrode used is 3 mm × 3 mm. To detect pulse waves before and after sweating, a human subject performed 10 min exercise by running up and down the stairs for perspiration. The capacitance of SEMS before and after exercise was recorded in Fig. 3g, and we found that the baseline capacitance recorded by SEMS electrode grew up from 1750 to 1790 pF as a result of increased sweat coverage on the skin (Fig. 3g). About 20 min later, the capacitance of the SEMS began to decrease and approached the initial capacitance within 20 min as sweat gradually evaporated. Such variance of baseline capacitance $C_0$ upon perspiration is expected, because numerous ions such as $K^+$, $Na^+$, and $Cl^-$ on the order of 10 mM in concentration in sweat will increase the ionic conductance of the corneum. Thus, when we put the SE and CE in conformal contact with the wet skin, the iontronic interface which separates electrons and ions may exhibit a higher capacitance than the case of dry skin. The gradual change in baseline is due to the different states of skin hydration. However, the change does not cause an adverse effect of detecting dynamical physiological signals such as pulse waves due to heartbeat. From eight data points collected in total (at 0, 12, 14, 16, 20, 30, 40, and 50 min), our experiment indicates that pulse waves can be continuously detected with a peak to valley amplitude of 0.96 ± 0.1 pF during the whole stage (Fig. 3h and Supplementary Fig. 13). The wave amplitude showed little change during the test, either with sweat or after the sweat is dried. The pulse rate (PR) could be extracted as well, which changes from 122 min$^{-1}$ right after exercise to 92 min$^{-1}$ at 50 min (Fig. 3h). Eventually, these data confirm that the capacitance baseline rises upon sweating while the amplitude of the physiological signals

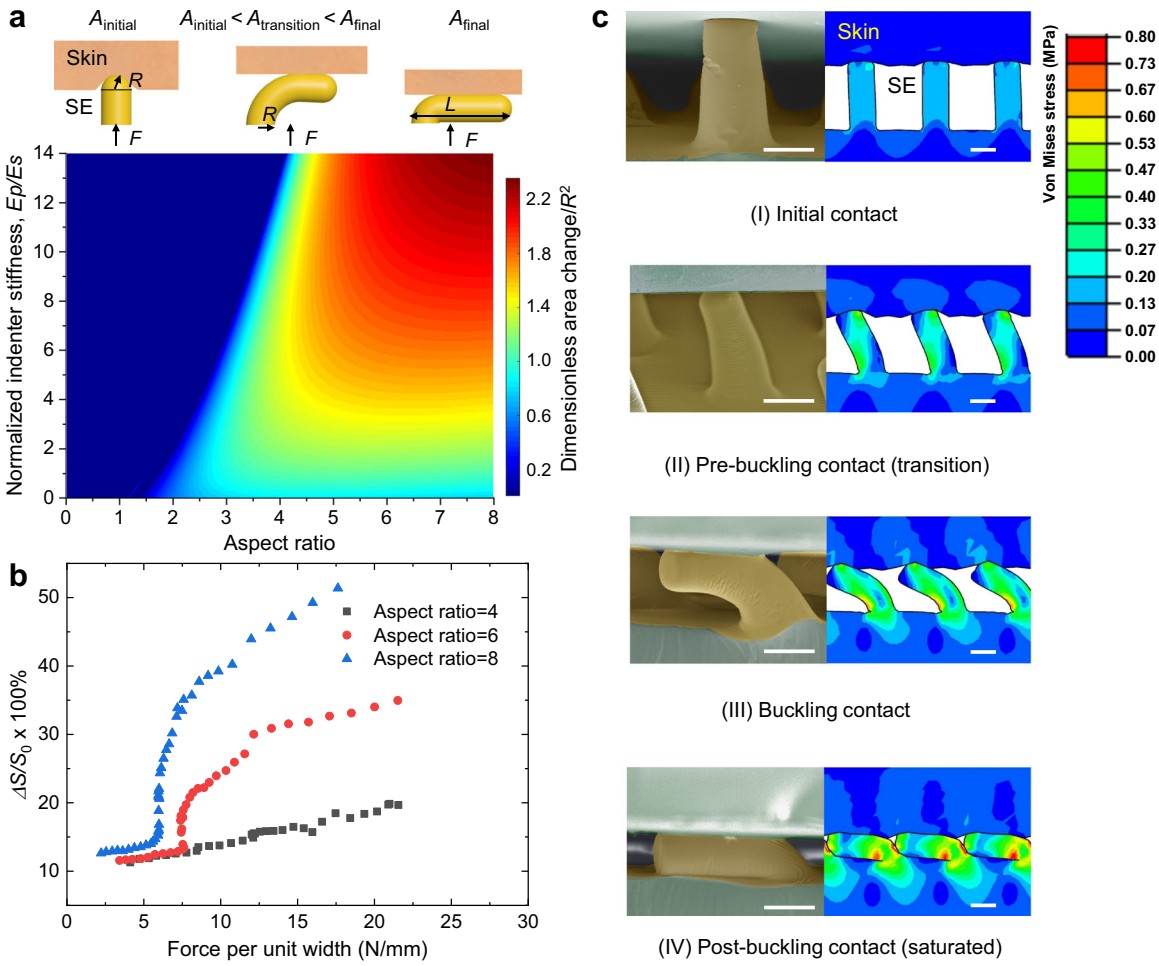

**Fig. 2 Structural design considerations and numerical simulation of compressive deformation of the SEMS. a** Schematics of elastic contact between a cylindrical pillar and a soft substrate before, during, and after the onset of buckling, here $A_{initial}$, $A_{transition}$, and $A_{final}$ indicate the contact areas of initial contact, transition, and post-buckling, respectively. The difference of areas between initial and final states normalized by the square of radius ($R^2$) are calculated via Hertz Contact Models around the point of Euler buckling load, and plotted onto the color map, with two coordinates being the aspect ratio of cylinder and the ratio of indenter stiffness ($E_p$) to that of the substrate ($E_s$). Deep blue region indicates the indenter is over-stiff such that Euler buckling is not likely within a reasonable range of skin strengths. **b** Finite element simulations of relative area change ($\Delta S$ normalized by the total available surface area $S_0$) under monotonically increasing loads for three different aspect ratios ($L/R$) of 4, 6, and 8. We assume plain strain condition with 33 pillars in a single unit. Boundary displacements are enforced via prescribing displacement and force feedbacks are averaged along the indenter's flat surface. **c** Visual comparison between experimental SEM images and FEA simulation results. The morphologies of deformed pillars throughout buckling show good consistency with experimental observations. Color indicates von Mises stress amplitudes. This experiment is repeated independently for at least three times (SE: sensing electrode). All scale bars are 10 μm.

(e.g., fingertip pulse waves) is not significantly affected. We also detected the response of the SEMS upon touch at different skin humidity and compared the signal intensity at the same applied forces (Fig. 3i). We found that the capacitance values with sweat under normal forces of 0.2, 0.4, and 0.6 N are higher than those before sweating, but the differences are limited (typically a few picofarads). In addition, the capacitance increased due to the improved skin humidity. Overall, sweating will increase the capacitance baseline but has little influence on the detection of both physiological and touching signals. Finally, we applied the sensing structure on the skin of different volunteer subjects from 6 to 60-year-old, and the results show that the SEMS could detect pulse waves of all subjects with limited change in wave amplitude (Supplementary Fig. 14).

Electrical noise is a challenge of high-quality signal recording for epidermal and wearable electronic sensors, and is dominantly attributed to the relative motion of the skin-electrode interface[15]. Here in this work on-body tests under various body motion modes and high-electrical noise environments were carried out to identify the effect of body motion and electrical interference on SEMS signal. In the experiment, the SE and the CE were attached on a fingertip and the ipsilateral forearm, respectively. The SEMS outputs stable fingertip pulse signals under quiet body conditions, without being affected by respiration or talking. By contrast, daily exercises including fast walking, slow walking, or waggling the hand with the SEMS generate obvious "interferences" to the pulse signal (Fig. 4a, and Supplementary Fig. 15). However, the signal amplitude from the motion artifacts (2–3 pF, see Fig. 4a) is far lower than that caused by touching (~hundreds of picofarads, see Fig. 3c). This illustrates a principle: the effect of motion-induced noise on tactile sensing is negligible. It is interesting that a further frequency-domain analysis based on Fourier and wavelet transformation of the signals well distinguishes the characteristic frequencies of the pulse and the characteristic frequency information of body motions (Fig. 4b). The wavelet transformation result indicates that the pulse signal and motion artifact information can be independently extracted without significant interference. In addition, the frequency analysis indicates that the

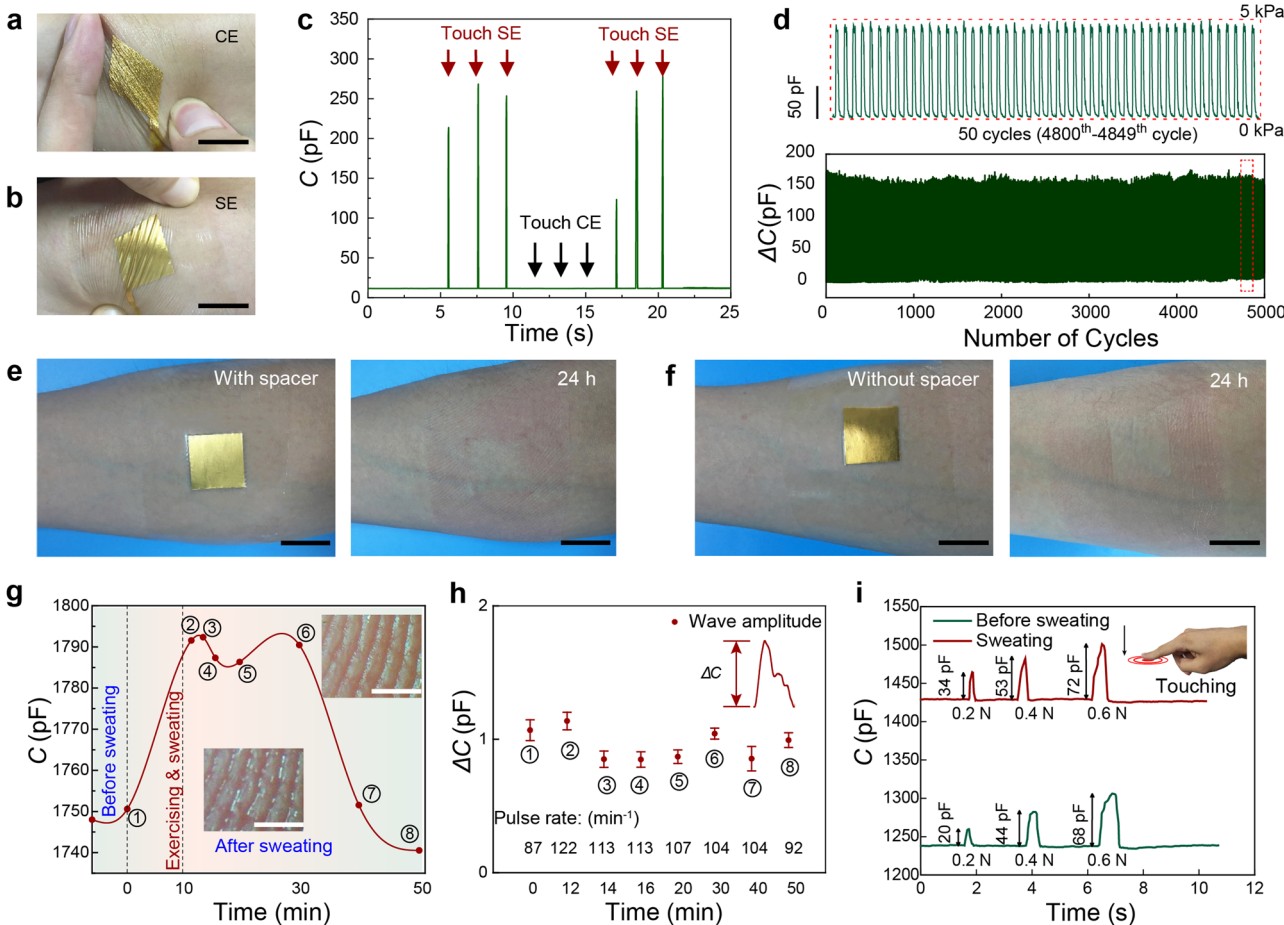

**Fig. 3 Stability of the SEMS and the influence of sweat on pulse and touch detection. a**, **b** Photographs of (**a**) the CE (counter electrode) and (**b**) SE (sensing electrode) under stretching (~10%). Scale bar: 2 cm. **c** Responses by touching the SE and CE. No output was found when touching the CE. **d** Signal stability over 5000 loading/unloading cycles at a peak pressure of 5 kPa. The picture on the top shows the details of 50 cycles (from the 4800th to the 4849th cycle). **e**, **f** Skin conditions after 24 h of attachment of the sensing electrode **e** with and **f** without a spacer. No skin irritation in the electrode area was observed during the attachment. Scale bar: 2 cm. **g** Recorded capacitance of the SEMS during the test from a human subject who performed 10 min of exercise and back to rest state for 40 min. The insets showed the optical photos of fingertip with sweat and no sweat. Scale bar: 0.5 mm. **h** The amplitudes of pulse waves at different points of 0, 12, 14, 16, 20, 30, 40, and 50 min. $n = 10$ consecutive measurements, each one represents value of amplitude of pulse wave. Data are presented as the mean values±standard deviations: 1.06 ± 0.07, 1.13 ± 0.06, 0.85 ± 0.06, 0.84 ± 0.05, 0.86 ± 0.05, 1.04 ± 0.04, 0.85 ± 0.09, and 0.99 ± 0.05 pF. The variation of pulse rates from 0 to 50 min is shown at the bottom. **i** The responses to touches before sweating and with sweating.

pulse frequency during body motions is higher than that in a stationary state, which can reasonably be attributed to the fact that body motion will cause immediate heart rate increase. As shown in Fig. 4c, the heart rate increases from 1.5 to 1.7 Hz as the body changes from a stationary state to fast walking with a frequency of ~0.8 Hz, and drops back to 1.5 Hz as the subject stops walking. Overall, the results on motion artifacts indicate that the SEMS has a high interference immunity in both tactile sensing and physiological signal detection, and is able to distinguish different body motion modes by analyzing the frequency information.

A clear advantage of such iontronic sensing devices is their immunity to high-frequency electromagnetic interference. Weak electromagnetic interference was found by approaching the SE or CE with a palm or a metal conductor at a distance ~1 cm, causing a tiny but detectable change in capacitance (<0.5 pF, as shown in Supplementary Fig. 16) because the fringing electric field around the SEMS is interfered by the approaching conductor. This effect, however, changes only the baseline of the signal slightly but shows no significant interference with the pulse signal or that caused by touching. Introducing other electromagnetic signals by

making a call with a cellphone (distance ~1 cm), or by exposing the SEMS to WiFi signals, or approaching the SEMS with a plastic bar does not cause detectable signal changes or any interferences.

**SEMS-based smart glove and its application in tactile perception.** Textile-based SEMS is a promising design that enables cost-effective and convenient wearables for tactile sensing of manipulation tasks for patients with tactile dysfunction (which is common in type 2 diabetes)[39], as well as for physiological detection. Textile is a more preferable selection of wearing than the PDMS pillar-based electrodes that has safely served human beings for thousands of years. Electrostatic flocking is a simple and cost-effective way to make hairy and slender textile structures, which can be used as high-aspect-ratio pillars in SEs. Here we designed a fully textile SEMS-based glove that consists of a perforated silk liner (hole diameter ~2.2 mm) serving as the spacer, a sensing layer of distributed SEs (diameter ~2 mm, aligned to the holes of the liner) adherent to carbon cloth threads, and a top protective textile layer on which the sensing layer is bonded (Fig. 5a). Each SE consists of thousands of electrostatically flocked PET pillars coated with a thin layer of Au film

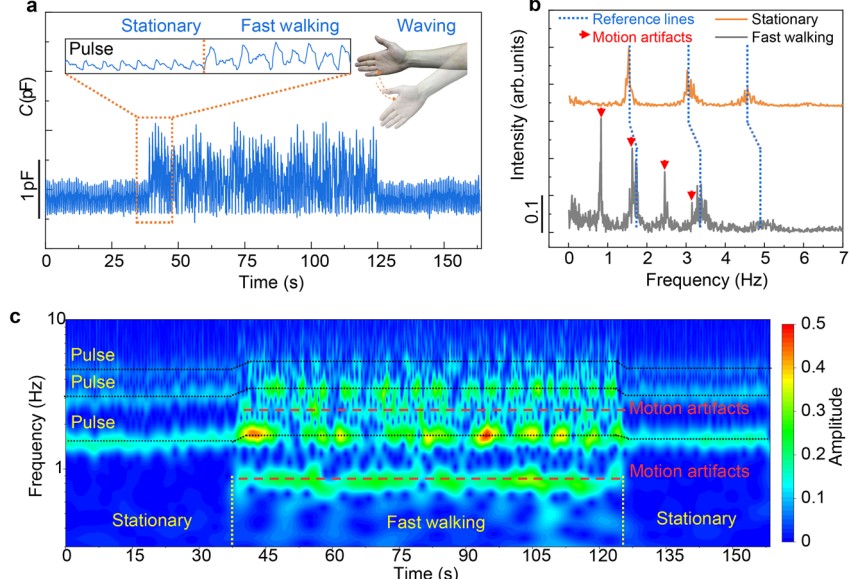

**Fig. 4 Interference immunity of electrical noise. a** Signal outputs during fast walking (walking frequency ~0.8 Hz), showing that sports cause limited noise amplitude. Tests were carried out with the SE attached on a thumb. Inset shows pulse signal and motion artifacts. **b** Frequency-domain analysis of the pulse signal with motion artifacts. **c** Continuous wavelet transform of the same signal in panel (**a**), revealing temporal dependence frequencies which coincide with the onset and termination of intense physical activity.

(Fig. 5b). The textile-based SE is non-irritable to the skin and can well keep the skin dry, and thereby exhibits negligible signal drift during long-term wearing. Because the flocked micropillars are mostly tilted (Fig. 5c), upon loading, the sensing elements exhibit a response without the "initial contact" but directly undergoing a buckling stage at which a maximal sensitivity is achieved, followed by a "post-buckling" stage (Fig. 5d and Supplementary Fig. 17) at which response becomes saturated. By contrast, a control textile sensing element with vertical and coarse flocked pillars (which are stiffer compared with the tilted and slender pillars) still exhibits an initial contact and pre-buckling stage (Supplementary Fig. 17a and c). Such experimental results well verify our modeling based on pillar buckling. Regarding sensing property, the textile-based sensing element performs similarly to the PDMS-micropillared sensing elements in terms of detecting pulse waves as well as motion mode analysis (Supplementary Fig. 18). In addition, the textile-based sensing element remains stable under repeated loading/unloading cycles (Supplementary Fig. 19), and shows no skin irritation or inflammation after 10 days of wearing on skin (Supplementary Fig. 20).

The SEMS-based smart glove contains 65 sensing elements covering all fingers and the palm with a higher SE density on fingertips than on the palm and the proximal segments of fingers, as shown in Fig. 5e. A shared CE is attached on the forearm. This smart glove can easily be worn on or taken off from hand. Before grasping task, the sensing elements have relatively loose contact with the skin and the capacitance values are low. Pressure mapping is conducted by recording the capacitance signal of all pixels when holding an object. It shows that grasp of a soft and compressible balloon causes a relatively uniform pressure distribution over the entire palm (Fig. 5f), while holding a hard beaker generates intensive signal amplitude on fingers but far weaker signal on the palm (Fig. 5g). The results can be reasonably ascribed to the fact that the contact between soft materials is more homogenous than that between hard materials, or between a hard and a soft material. Our glove well reflects the pressure distribution as well as the differences in rigidity of the tested objects. It is expected that this glove can be applied to identify the rigidity as well as the shape of various objects by applying big data

analysis and machine learning[40], being an ideal platform to help patients with haptic disability to recover their tactile capabilities for manipulation tasks and identification of objects. This wearable tactile platform may also be applied for health monitoring (such as blood pulse detection, Supplementary Fig. 18), motion detection, and human-machine interactions.

We also conducted experiments to measure the influence of perspiration on the tactile sensing performance of the SEMS-based smart glove. Signals of both pulse waves and touch were tested before and after sweating. In this specific experiment, the radius of each sensing element was 1.5 mm. After the human subject ran up and downstairs for 10 min, the variation of pulse waves and pulse rate (PR) was detected using the smart glove at eight time-intervals (at 0, 12, 14, 16, 20, 30, 40, and 50 min) (Supplementary Fig. 21). Figure 5h presents the capacitance recorded on one of the textile-based sensing electrodes placed on the fingertip of the glove during the test, and a change of baseline capacitance from 980 to 1120 pF is observed showing the effect of increasing sweat coverage, and gradually dropped to 990 pF in 40 min. Unlike the case of data taken on PDMS micropillar based sensing elements, we did not record any obvious plateau of baseline capacitance on the textile-based SEMS electrode during the change of skin hydration throughout the 50 min test period, and we attribute it to the fact that high gas permeability provided by the all-textile glove may facilitate evaporation of sweat into the ambient environment. Figure 5i presents that the valley to peak amplitudes (around ~1 pF) of the fingertip pulse waves experienced little change during the whole process, except for that at the beginning of perspiration (amplitude ~2.1 pF). Upon touching, the capacitance signal intensities of the sensing element subjected to 0.2, 0.4, and 0.6 N also show higher but very close values after sweating, together with an increased capacitance baseline (Fig. 5j). We further immersed the electrodes into simulated human sweat (300 mg NaCl dissolved in 100 g $H_2O$) for 12 h to test the effect of long-term sweating and the waterproof property of the electrodes. The results show that the function of the SEMS was not significantly affected after the electrodes were immersed in simulated sweat for 12 h (Supplementary Fig. 22). All these data indicate that our SEMS-based

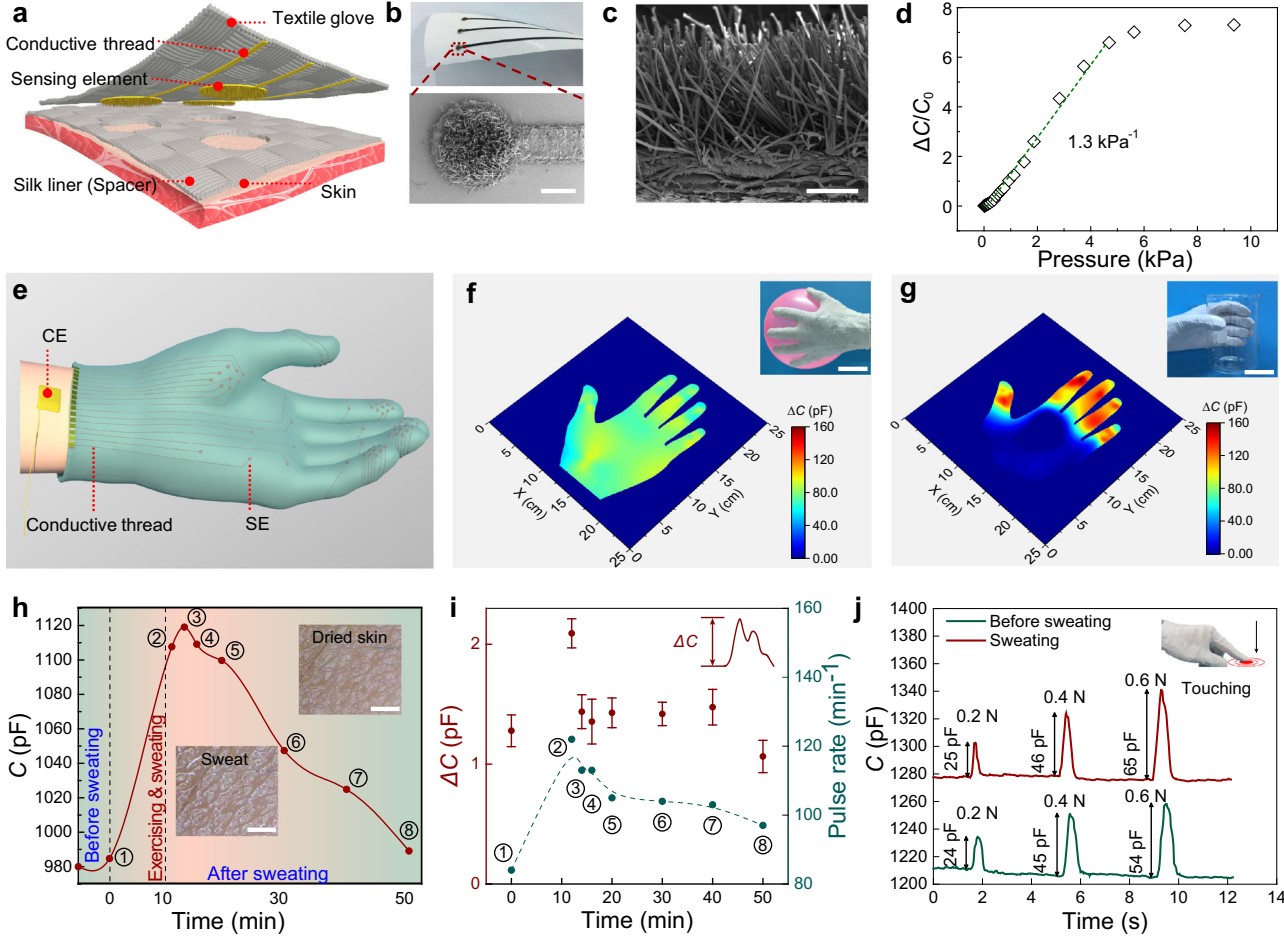

**Fig. 5 All-textile smart glove for pressure mapping and pulse wave detecting. a** Schematics of the fully textile sensing structure, showing a fully fabric structure that consists of a silk liner spacer, an electrode layer, and an outer protective glove. **b** Photograph of and low magnification SEM image of the SE used in the smart glove. Scale bar: 1 mm. **c** SEM image of the surface structures of the SE, showing tilted and slender pillars coated with a thin layer of Au film. The SEM observation is repeated independently for three times and similar results are presented. Scale bar: 200 μm. **d** Normalized change of capacitance as a function of applied pressure. **e** A three-dimensional schematic image of the smart glove. **f, g** Capacitance mapping of the smart glove worn on a healthy subject by holding **f** a balloon and **g** a beaker, indicating distinct pressure distributions for the two cases. Scale bar: 5 cm. **h** Recorded capacitance during the test from a stationary state to exercise (lasting for 10 min), and back to a rest state for 40 min. Insets show the skin with sweat and after the sweat is dried. Scale bar: 2 mm. **i** The amplitudes and rates of pulse waves at 0, 12, 14, 16, 20, 30, 40, and 50 min. $n = 10$ consecutive measurements, each one represents the value of amplitude of pulse wave. Data are presented as the mean values ± standard deviations: $1.27 \pm 0.13$, $2.09 \pm 0.12$, $1.43 \pm 0.14$, $1.35 \pm 0.18$, $1.42 \pm 0.12$, $1.41 \pm 0.09$, $1.47 \pm 0.14$, and $1.06 \pm 0.13$ pF. **j** The recorded SEMS sensor responses to touch before and with sweating.

smart glove is able to consistently detect physiological and pressure signals under different skin-hydration conditions.

## Discussion

The SEMS presents a few advantages in epidermal pressure sensing over existing e-skins. First, human skin is used to replace synthesized ionic materials or electronic active materials which may otherwise have biocompatibility and skin-irritation problems for on-skin applications. Ionic gels and hydrogels are commonly used as ionic materials for iontronic sensing. However, ionic gels are mostly toxic and water-absorbent[41], while hydrogels dehydrate easily in the air. Although Pan et al. first reported a skin-ionic gel interface for high-performance pressure sensing and skin was used as part of their sensor with an ion-ion interface[42,43], an artificial ionic gel playing as an active material is still required, whereas here the SEMS eliminates the need of any ionic liquid or gel.

The use of skin as the active material, on the other hand, significantly simplifies the sensor structure. Compared with existing epidermal pressure sensors or e-skins which often have a

sandwiched structure[7,8], the SEMS simply consists of two single-layered electrodes laminated on skin, while exhibiting far higher sensitivity and response speed. For example, the maximum sensitivity of our SEMS is about 83 times higher than that of a conformal nanomesh sensor[7], and the response speed is about 13 times higher. In addition, we have verified the iontronic nature of the SEMS and shown that the distance (from submillimeter to meter scale) between the CE and SE has little influence on its dynamic tactile sensing performance, and this allows the SEMS to be flexibly deployed on skin as epidermal devices or textile-based wearables.

Second, the SEMS presents low noise but high signal intensity upon touching because of its iontronic nature. We have shown that a submillimeter-sized sensing element exhibits a signal-to-noise ratio as high as 66 dB, well confirming the signal quality of the SEMS. Furthermore, since motion-related frequencies can be conveniently distinguished from the characteristic frequencies of pulse in the frequency domain, the extraction of physiological signal can be done without being affected by motion artifacts. We expect the device to find wearable applications to enable long-

term monitoring of patient vital signals such as pulse irregularity, analysis of motion modes for athletes, and recovery of sensing capability of people with tactile dysfunctions, and so forth. A portable system may further be integrated to enable continuous monitoring of body motion or touch without affecting the subjects' daily life activities (Supplementary Movie 2). In addition, biotissues other than skins can also form an iontronic interface with electrodes. As such, the sensing mode is also expected to be extended to other living systems, being applied in intelligent plants, pressure sensing for epidermal and implantable medical devices.

## Methods

**Fabrication of the micropillar sensing electrodes**. A silicon template with microholes (10 μm in diameter, 30 μm in depth, and 30 μm in pitch) was prepared in two steps. First, a gold disc array of tetragonal lattice (disc diameter: 10 μm, pitch: 30 μm) was made on a silicon wafer by photolithography followed by catalytic wet etching to form the hole array, to be used as a master template for the pillar formation. Next, the master template was treated by air plasma (TS-PL05, Dong Xin Gao Ke CO., Ltd) at 500 W for 10 min, and further surface-modified with 1H,1H,2H,2H-perfluorooctyltrichlorosilane (Macklin) in a vacuum desiccator for 24 h. Using the as-prepared silicon mold as the master template, a second polycarbonate (PC, Dongguan Ling Mei New Materials Co., Ltd) mold with micropillars was fabricated by using a hot embossing technique. Then, a mixture of Sylgard 184 base and curing agent in a weight ratio of 10:1 (Dow Corning Co., Ltd) was casted onto the PC mold, and cured at 70 °C for 3 h. After that, the cured PDMS layer with well-defined microholes was slowly peeled off from the PC mold, and was exposed to air plasma at 500 W for 5 min, followed by treating with 1H,1H,2H,2H-perfluorooctyltrichlorosilane in a vacuum desiccator for 24 h for better demolding.

Micropillared PDMS film was prepared by spin-coating PDMS solution onto the surface-treated PDMS mold at 800 rpm for 1 min. Then, the film was cured at 70 °C for 3 h and peeled off from the PDMS mold. Finally, a layer of Au film with a thickness of 100 nm was deposited onto the PDMS film by using electron-beam evaporation, forming a flexible electrode with micropillar arrays. The thickness of the Au film on the side surface of pillars was determined to be ~10 nm, judged from the period of surface wrinkles on the pillar side surface. A mask with patterned tetragonal array of holes (1.7 mm in diameter and 3.4 mm in hole pitch) and threads was used for depositing a layer of Au film with a thickness of 100 nm for the fabrication of micropillared electrode array. The electrodes were fixed on skin by using breathable 3 M tapes (Tegaderm film 1626 W), a layer of transparent dressing which shows gentle adhesive and high comfort on skin and forms a sterile barrier to external liquids, bacteria, and viruses. The fabrication of the micropillared electrodes is illustrated in Supplementary Fig. 23.

The spacer was made by laser-cutting a 38-μm-thick PET membrane with patterned microholes. For our sensing tests, spacers with tetragonal circular-hole array with a hole diameter of 1.45 mm and a hole fraction area of 66% were adopted.

**Fabrication of SEMS pixel array and the test of pressure mapping**. We used a mask for the fabrication of the sensing element array, with 1.7 mm in diameter and 3.4 mm in hole pitch. The experiment was conducted on a piece of porcine skin. The T-shaped rubber was cut using a blade, and then placed on the sensing elements array and the capacitance of each sensing element was measured.

**Fabrication of smart glove**. The textile glove, carbon cloth, and silk cloth were used as purchased. The conductive carbon cloth (from Shenzhen Meicheng Co., Ltd) was laser-cut into threads (0.8 mm in width) with a rounded end, which is 2.0 mm in diameter and consists of electrostatically flocked PET pillars that were coated with 100 nm Au by sputtering, serving as SEs. The conductive threads with the SEs were then bonded to the glove. The silk cloth (0.08 mm thickness) was cut into stripes with holes (2.2 mm in diameter) aligned to the sensing elements. The signal mapping was carried with 65 sensing elements; linear interpolation of the signals was used to make images smooth.

**Characterization and measurements**. The surface morphology of the sensing electrode was characterized by field-emission scanning electron microscopy (FE-SEM, TESCAN). The capacitance of the SEMS was measured using an LCR meter (E4980AL, KEYSIGHT) at a frequency of $10^5$ Hz, if not specified. The area of CE was 20 mm × 20 mm, and the area of SE for fingertip pulse detection was 5 mm × 5 mm, if not specified otherwise. A force gauge with a computer-controlled stage (XLD-20E, Jingkong Mechanical Testing Co., Ltd) was used to apply and record the external pressure loaded on the sensor. Porcine skin was used for the sensitivity tests. Before test, the porcine skin was immersed into 0.9% NaCl solution for 24 h at 2 °C.

The skin-irritation test was conducted by laminating the SE and CE on the skin of forearm of six subjects, the electrode-covered skins were observed after removing

the electrodes, and reports from the subjects was collected. The on-skin experiment was confirmed and approved by the IRB committee of the Southern University of Science and Technology (Nos. 20190007 and 20200031).

**Mechanical simulation**. Finite element analysis (FEA) was conducted using Abaqus/Explicit. We numerically simulated the displacement-controlled compression test of the SE. The scenario of PET spacer with a thickness of 38 μm and a percolated area ratio of 66% was used to match the experimental conditions. The skin and PDMS were modeled as linear elastic and incompressible neo-Hookean with Young's modulus $E_{skin}$ = 450 kPa and $E_{PDMS}$ = 2.0 MPa. PET spacers were assumed to be rigid. A compressive pressure of 20 kPa was applied on the top surface of the PDMS layer. All interfacial contacts were assumed to have a Coulomb friction coefficient of 0.4 to ensure numerical stability.

**Statistics and reproducibility**. All experiments were repeated independently with similar results for at least three times.

**Experiments on human subjects**. Informed consent was given by each human subject, or a parent of the subject (for the 6-year-old child), and all experiments were conducted under approval from the Institutional Review Board at the Southern University of Science and Technology under protocol number: 20200031.

**Reporting summary**. Further information on research design is available in the Nature Research Reporting Summary linked to this article.

## Data availability

All relevant data sets generated during and/or analyzed during the current study are available from the corresponding author upon request.

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

## Acknowledgements

The work conducted in SUSTech was funded by the National Natural Science Foundation of China (No. 52073138), the "Guangdong Innovative and Entrepreneurial Research Team Program" under contract No. 2016ZT06G587, the "Science Technology and Innovation Committee of Shenzhen Municipality" (Grant No. JCYJ20170817111714314), the "Guangdong Provincial Key Laboratory Program" (No. 2021B1212040001), and the Shenzhen Sci-Tech Fund (No. KYTDPT20181011104007). H.D. and N.X.F. acknowledge the financial support from SUSTech-MechE joint center. The authors thank Professor Zhigang Suo and Dr. Siya Huang for deep discussion.

## Author contributions

C.F.G. conceived the idea and designed the research. C.F.G. and N.F. directed the whole study. P.Z. and X.H. performed the majority of the experiments, P.L. designed the electrodes. P.Z., H.D., N.F., and C.F.G. analyzed the experimental data. H.D. and L.W. performed the finite elemental analysis, and H.D. and N.F. analyzed the deformation of the micro-electrodes and the sensing mechanism. H.D., N.F., P.Z., and C.F.G. analyzed the motion interference data. J.H., X.H., P.L., N.B., and Z.W. discussed the results and assisted in experimental design. X.H. and P.L. demonstrated the influence of perspiration on the SEMS. C.F.G., H.D., P.Z., and N.F. drafted the manuscript, and all authors contributed to the writing of the manuscript.

## Competing interests

The authors declare no competing interests.
