## [Peer Review File · Nature Communications]

REVIEWER COMMENTS

Reviewer #1 (Remarks to the Author):

The manuscript entitled "Skin-electrode Sensing Structure" introduced an iontronic pressure sensing structure of which human skin is used as one part of the sensing structure to achieve a simplified device architecture for pulse wave and motion detection. This concept is relatively new but not completely original, and therefore, the manuscript needs further improvement before its acceptance to NC.

1. The key innovation of this paper is claimed on that skin can be used as one component of the pressure sensing device, and thus, it can achieve a simplified sensing structure. Here, the interface between the skin, which can be treated as an ionic material for the sufficient electrolyte contained in the cell, and the electrode, either gold coated PDMS pillars or conductive fabric, will form a classic iontronic sensing structure. Based on the iontronic sensing mechanism, pressure applied to the structure will increase the physical contact area of such interface, resulting in the rise of the measured capacitance. The similar concept in fact was first introduced in several prior arts (Z. Zhu, R. Li, T. Pan, Adv. Mater. 2018, 1705122; Z. Zhu, R. Li, T. Pan, Haptics Symp. 2018, San Francisco, USA), which was not even included in the manuscript. In those papers, a flat ITO electrode membrane is directly attached on the skin with an air gap in between to form a similar pressure sensitive iontronic interface. The major difference lies on the selection on the electrode materials and deformation mode of the pillars. In brief, the major innovation, in which the skin serves as the functional component of the iontronic pressure sensor, needs to include proper citations and the prior arts needs to be carefully compared and discussed.

2. Since the major difference among this paper and the works listed in Q1 is the structure of the sensing electrode, the authors may want to emphasize on the difference and further investigate what advantages can be found by using such sensing electrode design. For instance, higher sensitivity, resolution or response rate? It is not very clear how the pillared sensing structure can achieve a better performance over the classic planar surfaces. Further parametric optimization needs to be included, from which desired geometrical design can be established.

3. Authors claimed a high spatial resolution using the SESS device, and built an array prototype in Figure 1f. How is this device amounted onto the skin? How was the T-shaped rubber image formed? In the Fig 1f it is difficult to visualize how it has been done on the skin amounted device.

4. The mechanical response of the sensor shown in figure 1d has four different slope rates, any advantages of such property can be found in real application? Or why the special surface structure, Au coated PDMS pillars, is needed in such architecture? Would the Au-coated surface fractured under the high strained PDMS deformation? How repeatable is such bending activities without special treatment on the metalized surfaces?

5. Furthermore, Au on PDMS always show extremely low adhesion, which may influence the repeatability of the sensor. Characterizations should be made to prove the high stability of the sensor under repeated load (at least 5000-10000 cycles) and methods should also be proposed to solve this problem.

6. It is shown that sweat can influence the output of the sensor, leading to the inaccuracy and drift of the output, most likely due to the change of the ionic concentration under various skin humidity conditions. In addition, skin temperature can also vary and affect the capacitive measurement. Authors need to consider a solution to address these standing issues, while using a living material.

7. Will the distance between the sensing electrode and the counter electrode influence the output of the sensor? The influential factors should be investigated and discussed.

Reviewer #2 (Remarks to the Author):

A high-quality paper should at least reflect two things: novelty and importance, and I am happy that I find both in this work. The authors depict a novel sensing mode based on the skin electrode interface. This sensing structure simply consists of a microstructured electrode and a conformable electrode attached on skin, which is different from any existing sensor configuration. Here, the human skin serves as the active material that forms an iontronic interface behaving like an electric double layer, and it eliminates the need of artificial electronic or ionic materials used in conventional e-skins and significantly simplifies sensor structure, while improving the sensing performances because the iontronic interface has a much higher specific capacitance than the common capacitive-type sensors. The authors also demonstrate the detection of tiny physiological signals such as fingertip pulse, and the pressure mapping during hand gripping using an all textile-based smart glove with microstructured electrodes adhered on the inner surface. It is interesting that the skin-electrode interface shows high interference immunity from both body motion and skin hydration. Owing to the innovation and its possible use in epidermal and wearable electronics, I enthusiastically recommend its publication in Nature Communications. I also have a few points that might help improve this work.

1. The authors claim that the sensing performance of the SESS is not sensitive to the distance between the two electrodes. Please explain.
2. I think the author may try an adhesive layer to replace the dressing layer used. Of course, this may not be done in this work, but it worth trying.
3. The authors should provide an IRB approval. I understand that the authors serve as the subject, but it is better if they can provide an IRB number.
4. There should be a scale bar for the photograph of Figure 1f.

Reviewer #3 (Remarks to the Author):

In this work, the authors present a simple mode of skin-electrode sensing structure that can provide the continuous, real-time variation in the electrical characteristics of the human skin, with capabilities of feeling touch and detecting physiological signal (fingertip pulse) under different skin humidity. Such device design can further enable pressure mapping with millimeter-scale spatial resolution in a fashion of SESS-based gloves. Overall, the paper layout constructs clear; however, the manuscript has a few limitations of importance that diminishes the overall value. I strongly suggest authors provide more detailed advances (not enough at current form) to allow readers to have better understanding.

Major:

1. First of all, the title of this paper, "Skin-electrode Sensing Structure", is overstatement and not clear. There are many types of skin-electrode sensing devices, including sweat sensing, hydration sensing (K. Kwon et al. PNAS 118, e2020398118, 2021), mechanical sensing such as skin-modulus measurements (C. Dagdeviren et al. Nat. Mater. 14, 728, 2015), etc. Instead of vague statement, the authors must give a specific, unique definition in the overall title of the work, distinguishing their work from other previous publications.
2. Particularly, a schematic illustration for the exploded view of the device layout structure should be given in the manuscript, with detailed dimension information such as size or thickness. Such information can present a better understanding for readers.
3. The authors claimed that their devices are wearable with a simple structure. However, the measurement also requires a benchtop system associated with capacitance meter or other external connections. Can authors comment on the possibility of implementing their technology as a complete wearable device to enable continuous monitoring without affecting the subjects' daily life activities?
4. Fig. 3 demonstrates signal recordings on forearm based on tactile sensing. The result itself is impressive. However, in addition to forearm or fingertip as shown in the manuscript, is the device response upon touch sensitive to other different skin site? For example, skin/tissue profiles vary a

- lot between different sites (e.g. hand joint, palm, etc). Can authors provide supplementary experiments for device response on other various sites of human body?
5. For the electrode performance of measurements, how about the measurement sensitivity (the signal variation of output capacitance) and measuring time (response time) compared to other similar work in this field for tactile and physiological sensing? Can authors highlight such information in the manuscript, as an emphasis for the biomedical application of their devices?
 6. Throughout the manuscript, device characteristics on curvy surface of skin are missing. Such discussion is particular of importance since the skin texture often presents a surface with different curvatures that typically vary among different locations.
 7. Can authors present some detailed information for their volunteer subjects for device measurements? Do these parameters affect the sensing results? In other words, are the signals different when the device is applied on different people (e.g. ages, skin hydration level, etc)?
 8. Figure 5 shows the electrodes can consistently detect physiological and pressure signals under different skin-hydration conditions. What is the waterproof property of the electrodes? For example, can authors show the device performance after complete immersion into human sweat solution within a specific duration (e.g. hours, days, etc)?
 9. I think directly evaporating Au on PDMS without other adhesive layer is hard to get a robust conductive layer, so I am doubt about the stability of Au layer on abraded and bended PDMS. And in Figure 5c, the nanopillars are so dense, how could Au be coated uniform on every dense nanopillar?
 10. In Figure S15, the electrode for skin irritation test is ultrasmall but the electrodes in main manuscript are much larger. Why you choose so small electrodes for this test?
 11. What's the contact impedance between electrodes and skin in different situations? Especially you even put a PET spacer between electrodes and skin.

Minor:

12. Figure S9 displays the capacitance signal when the SESS is attached on skin for 0 h and 24 h. Can authors give an explanation why the signal is consistently increased in respect to different frequencies after attaching on skin for 24 h (possibly due to the increase of skin humidity)? Will the signals continuously increase with further time of attaching?
13. Page 2 Lines 41-42, the authors stated that other tactile sensing methods such as piezoresistive devices suffer from sophisticated material synthesis protocols and the need of extra encapsulation to maintain the hydrated functional environment. Can the authors give some specific examples to illustrate it, in order to highlight their device advantages as a comparison?
14. Can authors clarify their material selection for elaboration of device performance as coupled with skin surface? For example, are there any specific reasons for using PDMS as sensor layer? Similarly, is there a specific reason that authors used Au as a metal layer?
15. The glove design in Figure 5 consists of 65 sensing elements. What is the specific resolution? Can the authors comment on the possibility of further increasing the spatial resolution for future use in real-world application?
16. What's the contact impedance between electrodes and skin in different situations? Especially you even put a PET spacer between electrodes and skin.
17. In Page 5, line 102, the gap is so small, especially for the dense nanopillar, do you have more experiments and data to prove the breathability?
18. In Page 15, line 313, The authors claim they replaced synthesized ionic materials by human skin, but the skin is still work there when other sensors adhere on skin. So I do not think this is a special point.
19. In Page 11, line 224, "By contrast, daily exercises including fast walking, slow walking, or waggling the hand with the SESS generate obvious "interferences" to the pulse signal". In Page 16, line 328, "since our SESS electrode is immune to noise artifacts under body motion". They are inconsistent.

REVIEWER COMMENTS AND RESPONSE FROM THE AUTHORS

Reviewer #1 (Remarks to the Author):

The manuscript entitled “Skin-electrode Sensing Structure” introduced an iontronic pressure sensing structure of which human skin is used as one part of the sensing structure to achieve a simplified device architecture for pulse wave and motion detection. This concept is relatively new but not completely original, and therefore, the manuscript needs further improvement before its acceptance to NC.

1. The key innovation of this paper is claimed on that skin can be used as one component of the pressure sensing device, and thus, it can achieve a simplified sensing structure. Here, the interface between the skin, which can be treated as an ionic material for the sufficient electrolyte contained in the cell, and the electrode, either gold coated PDMS pillars or conductive fabric, will form a classic iontronic sensing structure. Based on the iontronic sensing mechanism, pressure applied to the structure will increase the physical contact area of such interface, resulting in the rise of the measured capacitance. The similar concept in fact was first introduced in several prior arts (Z. Zhu, R. Li, T. Pan, Adv. Mater. 2018, 1705122; Z. Zhu, R. Li, T. Pan, Haptics Symp. 2018, San Francisco, USA), which was not even included in the manuscript. In those papers, a flat ITO electrode membrane is directly attached on the skin with an air gap in between to form a similar pressure sensitive iontronic interface. The major difference lies on the selection on the electrode materials and deformation mode of the pillars. In brief, the major innovation, in which the skin serves as the functional component of the iontronic pressure sensor, needs to include proper citations and the prior arts needs to be carefully compared and discussed.

Response: Thanks for pointing this out. Indeed, the human skin had been applied as a part of tactile sensor (Z. Zhu, R. Li, T. Pan, Adv. Mater. 2018, 1705122; Z. Zhu, R. Li, T. Pan, Haptics Symp. 2018, San Francisco, USA), which elaborately used a layer of iontronic film on top of ITO as the iontronic electrode. The recommended two references complemented our review of literature in the first section. In addition to recognizing the credits of earlier work based on a similar concept, we refined our expressions to emphasize on the simplicity and durability in our design and selection of material, which was more than enough to distinguish our work from previously published studies. First, the papers mentioned by the reviewer applied an ionic film to make contact with skin. Our work eliminated the need of any liquid or synthetic-gel environment which will likely induce biotoxicity and suffer from long-term stability unless additional encapsulating solution is available. Second, the sensor by Pan et al. was based on a skin-ionic layer interface, of which the charges were all ions. By contrast, our sensing structure was based on a skin-electrode interface, which was iontronic (with electrons and ions as charges).

We have added the two references (41,42) in the manuscript and made comparison with our work. Please see our revised text in Line 6-9, Page 16.

2. Since the major difference among this paper and the works listed in Q1 is the structure of the sensing electrode, the authors may want to emphasize on the difference and further investigate what advantages can be found by using such sensing electrode design. For instance, higher sensitivity, resolution or response rate? It is not very clear how the pillared sensing structure can achieve a

better performance over the classic planar surfaces. Further parametric optimization needs to be included, from which desired geometrical design can be established.

Response: Thanks for the suggestions. As mentioned in Point 1, the differences came from two aspects: First, our work was free of the use of ionic film, which otherwise requires additional packaging to make the sensor and could cause health problem to the skin. The elimination of the ionic film also simplified the sensing structure. Second, the sensor by Pan et al. was based on a skin-ionic layer interface, of which the charges are all ions. By contrast, our sensing structure was based on a skin-electrode interface, which was iontronic (with electrons and ions as charges). Their sensing mechanisms were different.

As for the structural design of sensing electrode, in the original manuscript, we had also conducted a parametric study of the effect of critical geometric factor (e.g., the aspect ratio of an individual pillar) on the sensor loading curve. As Fig. 2(b) suggested, the response curve exhibited a distinctive pattern change when the aspect ratio increased from 2 to 4, with extended steep regions when it was further increased. This provided guidance on optimizing the structural parameters to improve the sensor's performance metrics. Currently, we have a separate and ongoing work of the topological optimization of sensing element for desired properties, where the favorable geometry is obtained from scratch using several metrics as the objective functions (linearity and sensitivity among others), which shows promising progress but is out of the scope of the current work under consideration.

3. Authors claimed a high spatial resolution using the SESS device, and built an array prototype in Figure 1f. How is this device amounted onto the skin? How was the T-shaped rubber image formed? In the Fig 1f it is difficult to visualize how it has been done on the skin amounted device.

Response: The experiment was conducted on a piece of porcine skin. We used a mask for the fabrication of the sensing element array, with 1.7 mm in diameter and 3.4 mm in hole pitch. We cut the T-shaped rubber using a blade, placed it on the pixel array and measured the capacitance of each sensing element. The electrodes were fixed on skin using breathable 3M tapes (Tegaderm film 1626 W).

We have added the information in Methods section of the revised manuscript. Please see our revised manuscript from Line 20 Page 18 to Line 2 Page 19.

4. The mechanical response of the sensor shown in figure 1d has four different slope rates, any advantages of such property can be found in real application? Or why the special surface structure, Au coated PDMS pillars, is needed in such architecture? Would the Au-coated surface fracture under the high strained PDMS deformation? How repeatable is such bending activities without special treatment on the metalized surfaces?

Response: We chose the Au coated PDMS pillars as the electrode because such a structure exhibited desired biocompatibility and breathability. This is because the pillars allow for minimized initial contact with the skin, as well as the gaps between the pillars that help ventilate air and skin residues to keep the skin dry. On the other hand, the pillars also help achieving high sensitivity according to our simulation in Figure 2. In fact, the accuracy of recording in skin-interfaced capacitive sensor

could be significantly compromised by the presence of an epidermal barrier consisting of dead cell material (the stratum corneum). To compensate for this loss of sensitivity, we micro-patterned the substrate to introduce structures that not only increased the overall available contact area, hence the pressure sensing range, but also improved the sensitivity as buckled micro-posts induced rapid contact area change, which corresponded to the fast rising section of the loading curve in Fig. 2(b).

In a metal-elastomer bilayer, delocalized rupture will likely occur when subject to large deformations, no matter the interfacial adhesion is strong or not, because of the huge difference in elastic moduli between the gold film and PDMS that causes local interfacial debonding. The delocalized rupture, however, does not cause significant change in electrical conduction. In fact, inducing cracking is an important strategy toward highly stretchable electrodes—it is the formation of distributed cracks that guarantees the high stability of electrical conduction of the electrode. We have worked deeply on film cracking and interfacial adhesion (Guo et al. *Nature Communications* 2014, 5, 3121; *Nano Letters* 2016, 16, 594–600). In addition, strong adhesion may not help maintain electrical conduction—when the film is thin, strong interfacial adhesion will cause localized rupture of the film and thus the metal film will become non-conductive (Guo et al. *Nano Letters* 2016, 16, 594–600). Similar conclusions can also be drawn from Professor Zhigang Suo’s work (Suo et al. *Applied Physics Letters* 2006, 88, 204103). Therefore, the formation of cracks is not an issue to be concerned about, but an important mechanism that helps the device maintain stable working conditions under large and cyclic deformations.

Regarding the request from the reviewer, we have added the data of resistance under repeated bending/release for 5000 cycles. Please see our revised manuscript in Line 16-20 Page 9 and Supplementary Fig.11

5. Furthermore, Au on PDMS always show extremely low adhesion, which may influence the repeatability of the sensor. Characterizations should be made to prove the high stability of the sensor under repeated load (at least 5000-10000 cycles) and methods should also be proposed to solve this problem.

Response: Yes, adhesion affects not only the repeatability but also the flexibility of the electrode. Please see our response to Point 4 for the effect of adhesion in detail. Our new measurement suggested that the resistance only exhibited limited change upon cyclic deformation. Here, we also showed that the capacitance signal maintained stable under repeated loading/unloading for 5000 cycles with a peak pressure of 5 kPa (updated Figure 3d). The experiment was conducted using a porcine skin which was sealed with a piece of food wrap to avoid dehydration.

Please see our revised text in Line 11-17 Page 9, and updated Figure 3d.

6. It is shown that sweat can influence the output of the sensor, leading to the inaccuracy and drift of the output, most likely due to the change of the ionic concentration under various skin humidity conditions. In addition, skin temperature can also vary and affect the capacitive measurement. Authors need to consider a solution to address these standing issues, while using a living material.

Response: Indeed, the signal drift under different skin hydration conditions is a current limitation of our sensing device. Nevertheless, in Figure 3g-k and Figure 5h-j, we showed that for both pulse

signal measurement and pressure detection, sweat increased the capacitance baseline but had limited influence on the relative change in signal intensity within each physiological state (Figure 3 i,j). In fact, for many applications, we can just focus on the relative change in signal intensity instead of the absolute value. The comprehensive calibration of signal under various sweating conditions is also important yet complicated, and is out of the scope of this work.

7. Will the distance between the sensing electrode and the counter electrode influence the output of the sensor? The influential factors should be investigated and discussed.

Response: When we fixed the position of the sensing electrode, the distance between the two electrodes did not affect the output. The counter electrode is fully conformable to the skin texture and the skin-counter electrode interface has a constant and much larger capacitance (C_2) than that of the sensing electrode (C_1). Because of the iontronic nature of the interface, the capacitance is determined by the applied voltage (which is a constant during measurement) and the contact area between the electrodes and skin. The measured capacitance C roughly follows $1/C=1/C_1+1/C_2$. Since C_2 is far larger than C_1 , we get $C\sim C_1$. That is, the measured capacitance is predominantly determined by C_1 and has little dependence on the distance between the two electrodes.

Reviewer #2 (Remarks to the Author):

A high-quality paper should at least reflect two things: novelty and importance, and I am happy that I find both in this work. The authors depict a novel sensing mode based on the skin electrode interface. This sensing structure simply consists of a microstructured electrode and a conformable electrode attached on skin, which is different from any existing sensor configuration. Here, the human skin serves as the active material that forms an iontronic interface behaving like an electric double layer, and it eliminates the need of artificial electronic or ionic materials used in conventional e-skins and significantly simplifies sensor structure, while improving the sensing performances because the iontronic interface has a much higher specific capacitance than the common capacitive-type sensors. The authors also demonstrate the detection of tiny physiological signals such as fingertip pulse, and the pressure mapping during hand gripping using an all textile-based smart glove with

microstructured electrodes adhered on the inner surface. It is interesting that the skin-electrode interface shows high interference immunity from both body motion and skin hydration. Owing to the innovation and its possible use in epidermal and wearable electronics, I enthusiastically recommend its publication in Nature Communications. I also have a few points that might help improve this work.

Response: We very much appreciate the positive comments on our work.

1. The authors claim that the sensing performance of the SESS is not sensitive to the distance between the two electrodes. Please explain.

Response: This is related the iontronic nature of the skin-electrode interface. Please see our response to Point 7 of Reviewer #1 for a collective answer to a similar question raised by another reviewer.

2. I think the author may try an adhesive layer to replace the dressing layer used. Of course, this may not be done in this work, but it worth trying.

Response: Great point! Yes, an adhesive layer will help the fixation of the electrodes. We are actually working on this issue by using an electrical bioadhesive that binds the electrode and skin easily. We will report the results in a future work.

3. The authors should provide an IRB approval. I understand that the authors serve as the subject, but it is better if they can provide an IRB number.

Response: Thanks for pointing out the missing information. We have added the IRB number in the revised manuscript in Line 8-10 Page 20.

4. There should be a scale bar for the photograph of Figure 1f.

Response: Done as suggested.

Reviewer #3 (Remarks to the Author):

In this work, the authors present a simple mode of skin-electrode sensing structure that can provide the continuous, real-time variation in the electrical characteristics of the human skin, with capabilities of feeling touch and detecting physiological signal (fingertip pulse) under different skin humidity. Such device design can further enable pressure mapping with millimeter-scale spatial resolution in a fashion of SESS-based gloves. Overall, the paper layout constructs clear; however, the manuscript has a few limitations of importance that diminishes the overall value. I strongly suggest authors provide more detailed advances (not enough at current form) to allow readers to have better understanding.

Response: Thanks for the positive comments and the suggestion to improve this work.

Major:

1. First of all, the title of this paper, "Skin-electrode Sensing Structure", is overstatement and not clear. There are many types of skin-electrode sensing devices, including sweat sensing, hydration sensing (K. Kwon et al. PNAS 118, e2020398118, 2021), mechanical sensing such as skin-modulus measurements (C. Dagdeviren et al. Nat. Mater. 14, 728, 2015), etc. Instead of vague statement, the authors must give a specific, unique definition in the overall title of the work, distinguishing their work from other previous publications.

Response: Thanks for the suggestion. We have changed the title to be: Skin-Electrode Iontronic

Interface for Mechanosensing, to make the title more unique and specific.

2. Particularly, a schematic illustration for the exploded view of the device layout structure should be given in the manuscript, with detailed dimension information such as size or thickness. Such information can present a better understanding for readers.

Response: Thanks for the suggestion. We have added an exploded view of the device layout in Figure 1, which shows detailed dimensions of the device.

3. The authors claimed that their devices are wearable with a simple structure. However, the measurement also requires a benchtop system associated with capacitance meter or other external connections. Can authors comment on the possibility of implementing their technology as a complete wearable device to enable continuous monitoring without affecting the subjects' daily life activities?

Response: Thanks for the suggestion. We tried to make a portable testing system that can detect the capacitance signal. While the system is capable of detecting finger touch (Supplementary Movie S2), it is unable to discriminate the fine structures (the T- and P-waves) of the fingertip pulse. We therefore did not use the system in the manuscript. We have given comments in Line 5-7 Page 17 on this part.

4. Fig. 3 demonstrates signal recordings on forearm based on tactile sensing. The result itself is impressive. However, in addition to forearm or fingertip as shown in the manuscript, is the device response upon touch sensitive to other different skin site? For example, skin/tissue profiles vary a lot between different sites (*e.g.* hand joint, palm, *etc.*). Can authors provide supplementary experiments for device response on other various sites of human body?

Response: Thanks for the suggestion. Indeed, we have found that the output varies at different sites including fingers, palm, forearm, wrist, and finger joints. The information has been added in the supplementary information as suggested. Please see our revised manuscript in Line 7-9 Page 9 and Supplementary Fig.10.

5. For the electrode performance of measurements, how about the measurement sensitivity (the signal variation of output capacitance) and measuring time (response time) compared to other similar work in this field for tactile and physiological sensing? Can authors highlight such information in the manuscript, as an emphasis for the biomedical application of their devices?

Response: Great point! Indeed, our sensing structure presented higher sensitivity compared with most capacitive-type e-skins but lower than iontronic-type e-skins because there is a layer of dead cell congregate (stratum corneum) on the skin surface, compromising the performance of our sensor. The comparison in maximum sensitivity, response time, and normalized change in signal amplitude ($\Delta C/C_0$) with existing capacitive and iontronic types of sensors has been added in Supplementary Table S1. Accordingly, we have also modified the main text in Line 1-4, Page 7.

6. Throughout the manuscript, device characteristics on curvy surface of skin are missing. Such discussion is particular of importance since the skin texture often presents a surface with different curvatures that typically vary among different locations.

Response: In fact, the fingertip has a curved surface. Based on the suggestion of the reviewer, we have measured more data on the sensing performance by attaching the electrodes on the different parts of the human body, which have different curvatures. Please see our revised manuscript in Line 7-9 Page 9 and Supplementary Fig. 10.

7. Can authors present some detailed information for their volunteer subjects for device measurements? Do these parameters affect the sensing results? In other words, are the signals different when the device is applied on different people (e.g. ages, skin hydration level, etc)?

Response: We have added the detailed information of the subjects in the Methods section. We have also conducted measurements on subjects at different ages (including a 6-year-old girl and a 60-year-old male) in the Supplementary Information Fig. 13 and Line 12-14 Page 11 in our revised manuscript.

8. Figure 5 shows the electrodes can consistently detect physiological and pressure signals under different skin-hydration conditions. What is the waterproof property of the electrodes? For example, can authors show the device performance after complete immersion into human sweat solution within a specific duration (e.g. hours, days, etc)?

Response: Thanks for the suggestion. To test the effect of long-term sweating on the sensing performance of the SEMS, we have immersed our electrodes of SEMS-based smart glove into simulated sweat for 12 h. After that, the SEMS was tested by finger pressing and capacitance signal was recorded accordingly. The data are shown in Supplementary Fig. 21, indicating that the SEMS remains stable after long-term sweating.

Please see our revised manuscript in Line 16-20, Page 15.

9. I think directly evaporating Au on PDMS without other adhesive layer is hard to get a robust conductive layer, so I am doubt about the stability of Au layer on abraded and bended PDMS. And in Figure 5c, the nanopillars are so dense, how could Au be coated uniform on every dense nanopillar?

Response: Thanks for pointing out the adhesion problem. Indeed, the adhesion is very important for the robustness of the sensing structure. From our experimental results (updated Figure 3d and Figure S11), both the electrode and the device exhibited acceptable stability. In fact, although there were cracks formed on the Au film, it still exhibited quite good electrical conductance due to delocalized rupture, and the cracked Au film was still well interconnected. Detailed interpretation can be found in our response to Point 4 and 5 of Reviewer #1.

In Figure 5c, the Au film was deposited by sputtering, which has a lower shading effect than e-beam evaporation or thermal evaporation. Of course, the film thickness on the pillar surface was not uniform. However, the uniformity had little influence on the sensing performance since capacitance signal was not sensitive to the electrical conductance of the electrodes.

10. In Figure S15, the electrode for skin irritation test is ultrasmall but the electrodes in main manuscript are much larger. Why you choose so small electrodes for this test?

Response: We used a small area electrode for the skin irritation test, because that was actually a sensing element of the smart glove (the same as what is shown in Figure 5b).

11. What's the contact impedance between electrodes and skin in different situations? Especially you even put a PET spacer between electrodes and skin.

Response: Thanks for the suggestion. We have measured the contact impedances of both the counter electrode (CE), and the sensing electrode (SE) with and without a spacer. Data are shown in Figure S4 and we also add the relevant content in our revised manuscript in Line 12-15, Page 6.

Minor:

12. Figure S9 displays the capacitance signal when the SESS is attached on skin for 0 h and 24 h. Can authors give an explanation why the signal is consistently increased in respect to different frequencies after attaching on skin for 24 h (possibly due to the increase of skin humidity)? Will the signals continuously increase with further time of attaching?

Response: Yes, the increased signal intensity was caused by the increase of skin humidity. Signal did not increase after 24 h due to saturation.

13. Page 2 Lines 41-42, the authors stated that other tactile sensing methods such as piezoresistive devices suffer from sophisticated material synthesis protocols and the need of extra encapsulation to maintain the hydrated functional environment. Can the authors give some specific examples to illustrate it, in order to highlight their device advantages as a comparison?

Response: A specific example has been added in the Introduction. Please see the added text in Line 16-19, Page 2, as well as newly added references (Ref. 14) in the revised manuscript.

14. Can authors clarify their material selection for elaboration of device performance as coupled with skin surface? For example, are there any specific reasons for using PDMS as sensor layer? Similarly, is there a specific reason that authors used Au as a metal layer?

Response: We used PDMS because it is soft (which facilitates the initiation of buckling mode and allows for high sensitivity) and biocompatible. We used Au for metallization because Au has very good compatibility.

These reasons have also been added in the revised manuscript in Line 9, Page 5.

15. The glove design in Figure 5 consists of 65 sensing elements. What is the specific resolution? Can the authors comment on the possibility of further increasing the spatial resolution for future use in real-world application?

Response: The resolution at fingertips was about 6 mm. Further improvement of spatial resolution is possible by adding more sensing elements (see Supplementary Fig. 8). However, the increasing number of pixels may lead to significant challenges in wiring and testing.

16. What's the contact impedance between electrodes and skin in different situations? Especially you even put a PET spacer between electrodes and skin.

Response: We refer the reviewer to Point 11 for our combined answer to this question.

17. In Page 5, line 102, the gap is so small, especially for the dense nanopillar, do you have more experiments and data to prove the breathability?

Response: Breathability plays an important role for the applications of the SEMS. The high stability of the pillar structures has been proven by Bae et al. (*Adv. Healthc. Mater.* 2013, 2, 109–113). Probably we can make sparse pillars to further improve breathability, but this leads to lower signal since signal intensity is determined by the contact area.

18. In Page 15, line 313, The authors claim they replaced synthesized ionic materials by human skin, but the skin is still work there when other sensors adhere on skin. So I do not think this is a special point.

Response: In the current form of most electronic skins, a piezoresistive, dielectric, or piezoelectric layer is sandwiched between a layered structure of two electrodes, and the skin underneath only serves as an object where measurement is conducted upon. It does not directly participate in the ion/electron transport processes. In those sensory devices, we could hypothetically replace the skin with another non-conductive material (like a timber), and they would still function properly. On the other hand, in our design, human skin actively engages in the exchange of ionic signal, where free electrons serve as the carriers in the electrodes while ionic fluxes contribute to the conduction in the tissue (see line 12-14 on Page 3). Not only is the sensor construction greatly simplified by removal of synthetic ionic material, but the whole device is much more stable against dehydration. In precis, these are the reasons why we emphasize that using skin as a functional component of the iontronic sensing device is an essential part of this work.

19. In Page 11, line 224, “By contrast, daily exercises including fast walking, slow walking, or wagging the hand with the SESS generate obvious “interferences” to the pulse signal”. In Page 16, line 328, “since our SESS electrode is immune to noise artifacts under body motion”. They are inconsistent.

Response: Sorry for the misleading information. Exercise will cause change of waveform of the

pulse in time-domain. This is because motion also possesses its own characteristic signal pattern. Thus, in the frequency domain it can be conveniently distinguished from the characteristic frequencies of pulse, and the extraction of physiological signal can be done without being affected by motion artifacts. Details are seen in Figure 4.

We have rewritten this part in order to make it clear in Line 1-3, Page 17.

REVIEWERS' COMMENTS

Reviewer #1 (Remarks to the Author):

In the revised manuscript "Skin-Electrode Iontronic Interface for Mechano sensing" authors have sufficiently answered most of the questions we suggested before, and presented the innovations and differences compared with the current technologies. However, additional minor modifications about question 3 and question 4 should be amended. In Q3, the authors claimed that the SESS device was placed on a piece of porcine skin, and the tactile image was then recorded, while figure 1f still has not represented this experiment design. The photos, which contains all important elements for this experiment, should be illustrated, or the audience could be confused with this demonstration. In Q4, the reason to choose Au coated PDMS pillars over current materials and design was attributed to the biocompatibility and breathability. These explanations should be added into the main text for the comparison, and the citations proving the biocompatibility and breathability of the new design should also be added.

In conclusion, the revised manuscript has reached the acceptable level of Nature Communications, and no additional review is needed if the authors have completely addressed the issues here.

Reviewer #2 (Remarks to the Author):

The authors have satisfactorily answered all the questions and the paper can be accepted.

Reviewer #3 (Remarks to the Author):

The authors have addressed all my concerns, and the paper's quality has been improved significantly. I believed this format can be published in Nat Comm as is.

AUTHOR'S RESPONSE TO REVIEWERS

REVIEWERS' COMMENTS

Reviewer #1 (Remarks to the Author):

In the revised manuscript “Skin-Electrode Iontronic Interface for Mechanosensing” authors have sufficiently answered most of the questions we suggested before, and presented the innovations and differences compared with the current technologies. However, additional minor modifications about question 3 and question 4 should be amended. In Q3, the authors claimed that the SESS device was placed on a piece of porcine skin, and the tactile image was then recorded, while figure 1f still has not represented this experiment design. The photos, which contains all important elements for this experiment, should be illustrated, or the audience could be confused with this demonstration. In Q4, the reason to choose Au coated PDMS pillars over current materials and design was attributed to the biocompatibility and breathability. These explanations should be added into the main text for the comparison, and the citations proving the biocompatibility and breathability of the new design should also be added.

In conclusion, the revised manuscript has reached the acceptable level of Nature Communications, and no additional review is needed if the authors have completely addressed the issues here.

Response: Thanks for the feedback and the additional comments. We have addressed these two points in the updated main text. Specifically, new captions were added on top of Figure 1f to remove any obscurity. We also introduced a new sentence with references to two papers confirming the biocompatibility and breathability of PDMS (See page 5, line 20). Once again, we would like to extend our thanks to the reviewer for the effort and time they put into polishing the manuscript.

Reviewer #2 (Remarks to the Author):

The authors have satisfactorily answered all the questions and the paper can be accepted.

Response: We would like to extend our thanks to the reviewer again for the effort and time they put into polishing the manuscript.

Reviewer #3 (Remarks to the Author):

The authors have addressed all my concerns, and the paper's quality has been improved significantly. I believed this format can be published in Nat Comm as is.

Response: We would like to extend our thanks to the reviewer again for the effort and time they put into polishing the manuscript.